# MDReID: Modality-Decoupled Learning for Any-to-Any Multi-Modal Object Re-Identification

**Yingying Feng**[1][*], **Jie Li**[2][*], **Jie Hu**[3], **Yukang Zhang**[2],
**Lei Tan**[3][†], **Jiayi Ji**[2,3][‡]
[1]Northeastern University [2]Xiamen University [3]National University of Singapore
fengyingying@stumail.neu.edu.cn, tanlei@nus.edu.sg

## Abstract

Real-world object re-identification (ReID) systems often face modality inconsistencies, where query and gallery images come from different sensors (e.g., RGB, NIR, TIR). However, most existing methods assume modality-matched conditions, which limits their robustness and scalability in practical applications. To address this challenge, we propose MDReID, a flexible any-to-any image-level ReID framework designed to operate under both modality-matched and modality-mismatched scenarios. MDReID builds on the insight that modality information can be decomposed into two components: modality-shared features that are predictable and transferable, and modality-specific features that capture unique, modality-dependent characteristics. To effectively leverage this, MDReID introduces two key components: the Modality Decoupling Learning (MDL) and Modality-aware Metric Learning (MML). Specifically, MDL explicitly decomposes modality features into modality-shared and modality-specific representations, enabling effective retrieval in both modality-aligned and mismatched scenarios. MML, a tailored metric learning strategy, further enforces orthogonality and complementarity between the two components to enhance discriminative power across modalities. Extensive experiments conducted on three challenging multi-modality ReID benchmarks (RGBNT201, RGBNT100, MSVR310) consistently demonstrate the superiority of MDReID. Notably, MDReID achieves significant mAP improvements of 9.8%, 3.0%, and 11.5% in general modality-matched scenarios, and average gains of 3.4%, 11.8%, and 10.9% in modality-mismatched scenarios, respectively. The code is available at: https://github.com/stone96123/MDReID.

## 1 Introduction

Object Re-Identification (ReID) focuses on identifying and retrieving specific objects across non-overlapping camera views. Conventional object re-identification (ReID) methods [1, 2, 3, 4, 5, 6, 7] primarily rely on RGB images. However, RGB-based approaches often face significant limitations under challenging environmental conditions, such as poor illumination, shadows, and low image resolution. These adverse conditions frequently lead to the extraction of misleading features, thereby diminishing discriminative capability. To overcome these limitations, multi-modal ReID [8, 9, 10, 11, 12, 13], which integrates complementary information from multiple spectrums, has emerged as a promising approach. By leveraging the inherent strengths and complementary features of different modalities, multi-modal ReID significantly enhances feature representations, enabling more robust and accurate identification in complex real-world scenarios.

---

[*]These authors contributed equally to this work.
[†]Corresponding Author: Lei Tan
[‡]Project Leader

39th Conference on Neural Information Processing Systems (NeurIPS 2025).

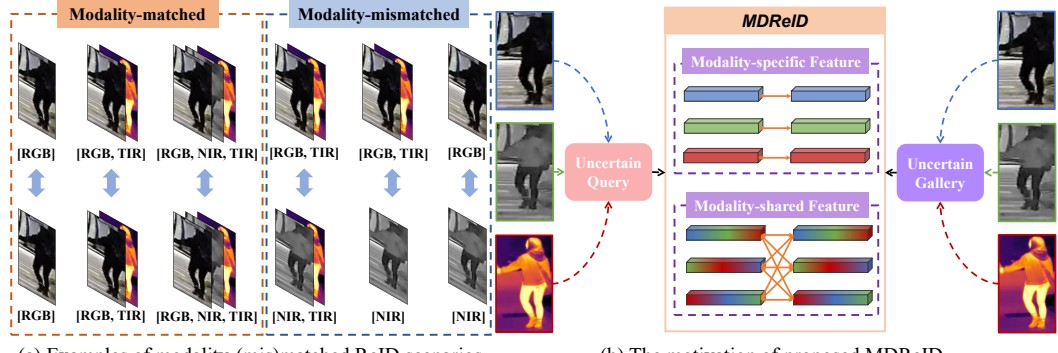

(a) Examples of modality-(mis)matched ReID scenarios.

(b) The motivation of proposed MDReID.

Figure 1: **Illustration of the motivation of MDReID. (a)** Though the availability of spectral modalities (e.g., RGB, NIR, TIR) varies across queries and galleries, recent methods only focus on the modality-matched scenarios, which limits their practical applicability. **(b)** MDReID overcomes the rigidity of modality constraints by disentangling modality-shared and modality-specific features, enabling effective matching between queries and galleries from arbitrary modalities.

The introduction of multi-modal data has significantly enhanced the performance of ReID in challenging scenarios, thereby stimulating extensive research efforts in this field [8, 9, 10, 11, 12]. EDITOR [11] selects diverse tokens from Vision Transformer (ViT) to mitigate the impact of irrelevant backgrounds and narrow the gap between modalities, achieving strong detection performance. TOP-ReID [12] incorporates a cyclic token permutation module to aggregate multi-spectral features and a complementary reconstruction module to minimize the distribution gap across different image spectra. These innovations enable TOP-ReID to achieve robust performance under both modality-complete and modality-missing scenarios. Although these methods effectively leverage the characteristics of multi-modal data, they are developed based on a fundamental assumption that all modalities within the dataset are strictly aligned (as illustrated in Figure 1 (a) Modality-matched). Ideally, every camera would simultaneously provide RGB, NIR, and TIR modalities. However, due to practical constraints such as device differences and deployment diversity, achieving fully matched retrieval conditions is challenging in real-world scenarios. Therefore, it is crucial to develop a flexible image-level any-to-any ReID framework capable of effectively operating under both modality-matched and modality-mismatched scenarios (as illustrated in Figure 1 (a)).

To address this limitation, we propose MDReID, a flexible any-to-any ReID framework designed to support retrieval tasks involving arbitrary combinations of query and gallery modalities. The core challenge of this problem lies in learning effective representations that remain robust under both modality-matched and modality-mismatched conditions. A straightforward solution, as adopted by TOP-ReID [12], attempts to predict the missing modality representation from the available one, thereby enabling modality-aligned retrieval. However, as highlighted by RLE [14], predicting cross-spectrum features solely from visual inputs constitutes an ill-posed problem. The modality-specific characteristics that are inherently unpredictable often lead to suboptimal learning. To this end, MDReID introduces a Modality-Decoupled Learning (MDL), which introduces a modality-shared and modality-specific token into the ViT architecture, leveraging multi-layer attention within the transformer to extract shared and specific features from multi-modal inputs. MDL explicitly disentangles the modality representations into two complementary components: a modality-shared component that captures the predictable cross-modal representation and a modality-specific component that preserves the unpredictable modality characteristics. This decoupling enables the model to better generalize across diverse modality combinations while maintaining the modality-specific cues.

Furthermore, to enhance the disentanglement between the two components, we propose a Modality-aware Metric Learning (MML) strategy. MML comprises two complementary objectives: a representation orthogonality loss (ROL) and a knowledge discrepancy loss (KDL). ROL is applied at the channel level to both promote the aggregation of modality-shared features across modalities and enforce orthogonality between shared and specific components, ensuring they capture distinct and non-overlapping information. KDL, on the other hand, encourages representational complementarity between shared and specific components by ensuring that the combined representation is more discriminative than either component alone.

To sum up, the main contributions of this paper are as follows:

- We propose MDReID, a flexible any-to-any object re-identification framework that supports retrieval across arbitrary query-gallery modality combinations, addressing the practical limitations of strictly aligned multi-modal datasets.

- We introduce a Modality-Decoupled Learning (MDL) strategy, coupled with a Modality-aware Metric Learning (MML) strategy. MDL explicitly disentangles modality-shared and modality-specific representations, while MML further strengthens this disentanglement by introducing a representation orthogonality loss and a knowledge discrepancy loss, encouraging the two components to encode distinct and complementary information.

- Extensive experiments on multi-spectral object ReID datasets (RGBNT201, RGBNT100, MSVR310) demonstrate the superior adaptability and performance of the MDReID across diverse scenarios. MDReID achieves significant mAP improvements of 9.8%, 3.0%, and 11.5% in general modality-matched scenarios, and average gains of 3.4%, 11.8%, and 10.9% in modality-mismatched scenarios, respectively.

## 2 Related Work

Compared to the general RGB-to-RGB object re-identification [15, 16, 17, 18, 19, 20, 21], multi-modal object re-identification (ReID) has demonstrated significant practical value, achieving widespread applications in cross-scenario systems [22, 23] (e.g., security surveillance, intelligent transportation, etc.). Current methods [24, 9, 25] primarily enhance re-identification accuracy by effectively leveraging the complementarity and integration of multi-modal features. For example, Zheng et al. [26] proposed the cross-directional consistency network (CCNet), which overcomes discrepancies in both modality and sample aspects, thereby achieving more effective multi-modal data collaboration. To address the modal-missing problem, Zheng et al. [27] proposed the DENet model, which incorporates a feature transformation module to recover information from missing modalities and a dynamic enhancement module to improve multi-modality representation. Li et al. [8] proposed HAMNet, which automatically fuses different spectral features using a specially designed heterogeneous score coherence loss. He et al. [28] proposed the GPFNet model, which utilizes graph learning to fuse multi-modal features.

Notably, models based on Vision Transformer (ViT) [29, 30, 10, 31, 32, 33] for object re-identification have achieved excellent results in recent years and gained widespread attention. Pan et al. [34] proposed the H-ViT model, which alleviates challenges caused by heterogeneous multi-modalities and reduces feature deviations arising from modal variations. Additionally, they introduced a progressively hybrid transformer (PHT [35]) that effectively fuses multi-modal complementary information through random hybrid augmentation and a feature hybrid mechanism. Crawford et al. [36] addressed the issue of modality laziness in multi-modal fusion by proposing the UniCat model based on a ViT architecture. By selecting diverse tokens from ViT to mitigate the impact of irrelevant backgrounds and narrow the gap between modalities, Zhang et al. [11] developed the EDITOR model. Wang et al. [12] designed the Top-ReID model, which achieves state-of-the-art accuracy by incorporating a cyclic token permutation module to aggregate multi-spectral features and a complementary reconstruction module to minimize the distribution gap across different image spectra.

Despite advancements in multi-modal object ReID, existing methods rely on modality-aligned conditions, limiting their adaptability to real-world scenarios. To address this, we introduce MDReID, which disentangles modality-shared and modality-specific features for effective retrieval across both matched and mismatched conditions. It incorporates Modality-Decoupled Learning (MDL) to refine feature separation and Modality-aware Metric Learning (MML) to enhance discrimination through orthogonality and knowledge discrepancy losses. As a result, MDReID enables robust object re-identification across diverse modality configurations.

## 3 Methodology

### 3.1 MDReID: Any-to-any Object ReID

In ideal multispectral person re-identification (ReID) settings, complete RGB, NIR, and TIR modalities are available for both query and gallery (RNT-to-RNT). However, real-world deployments often suffer from different modalities due to heterogeneous sensor availability and deployment

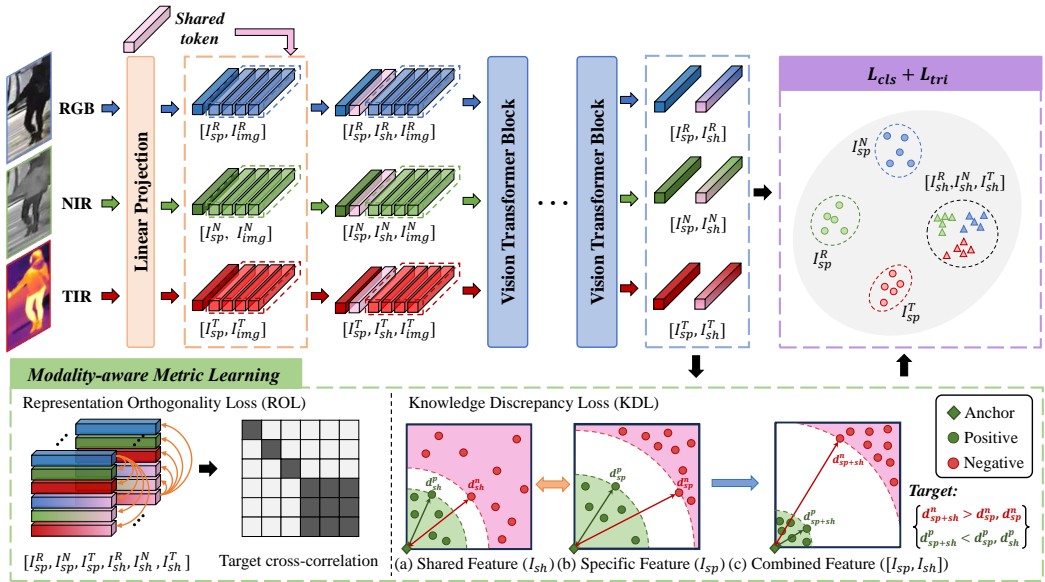

Figure 2: **Overall framework of MDReID.** MDReID is designed to support retrieval across arbitrary modality combinations. It disentangles features into shared and specific components to boost performance in both matched and mismatched scenarios. Additionally, by leveraging representation orthogonality loss (ROL) and knowledge discrepancy loss (KDL), MDReID refines feature separation and enhances retrieval robustness.

constraints. To address this fundamental challenge, we propose MDReID, a flexible, any-to-any ReID framework that supports arbitrary combinations of input and output modalities. As illustrated in Fig. 2, MDReID adopts a Vision Transformer (ViT) backbone and integrates two core components: Modality-Decoupled Learning (MDL) and Modality-Aware Metric Learning (MML). MDL explicitly disentangles each modality's representation into modality-shared features (crucial for cross-modality retrieval) and modality-specific features (essential for preserving discriminative cues in matched scenarios). MML further enhances this decoupling by enforcing metric-level consistency across modalities. This design enables robust and flexible object re-identification in diverse and modality-incomplete environments, significantly advancing the applicability of ReID in practical settings.

## 3.2 Modality Decoupled Learning

Generally, with a ViT as the backbone, we process input images from different modalities ($M \in \{R, N, T\}$, corresponding to RGB, NIR, and TIR). Standard ViT procedure involves segmenting the image into non-overlapping patches, which are then linearly projected into patch embeddings as:

$$I^M = \{[I_{\text{cls}}^M], I_1^M, I_2^M, \ldots, I_n^M\}, \quad \text{where } M \in \{R, N, T\}. \tag{1}$$

To explicitly model and decouple modality characteristics crucial for cross-modal ReID, we deviate from the standard use of a single [CLS] token. Instead, we prepend two distinct learnable tokens to the patch embedding sequence for each modality: a modality-specific token and a modality-shared token. This augmented sequence is then processed by the ViT encoder to get:

$$I_e^M = \{[I_{\text{sp}}^M, I_{\text{sh}}^M], I_1^M, I_2^M, \ldots, I_n^M\}, \quad \text{where } M \in \{R, N, T\}. \tag{2}$$

where $I_{sp}^M$ denotes the modality-specific features, $I_{sh}^M$ denotes the modality-shared features, and $I_e^M$ represents the encoded features.

Leveraging these decoupled features from different modalities ($I_{sp}^R, I_{sp}^N, I_{sp}^T, I_{sh}^R, I_{sh}^N, I_{sh}^T$), we construct a unified representation for each sample using a fixed-dimensional structure and an associated availability mask. This allows consistent handling across different modality presence scenarios. Specifically, we define a potential full feature vector, $v_{full}$, by concatenating all possible specific and shared components, assuming all three modalities (RGB, NIR, TIR) are present:

$$v_{full} = \left[ I_{\text{sp}}^R, I_{\text{sp}}^N, I_{\text{sp}}^T, I_{\text{sh}}^R, I_{\text{sh}}^N, I_{\text{sh}}^T \right]. \tag{3}$$

Correspondingly, a binary availability mask, $mask_{full}$, indicates the presence of each component:

$$mask_{full} = [1, 1, 1, 1, 1, 1], \tag{4}$$

where '1' signifies that the feature component at that position is present and valid, and '0' (as introduced below) signifies absence or invalidity.

The actual feature vector $v$ and mask $mask$ used for any given input sample are derived from this full structure based on the modalities available for that specific sample. If a modality $M \in \{R, N, T\}$ is missing for a sample, the corresponding modality-specific feature $I_{sp}^M$ and the modality-shared feature $I_{sh}^M$ in the vector $v$ are replaced with a zero vector $\mathbf{0}$. Also, the entries in the mask $mask$ corresponding to these zeroed-out features ($I_{sp}^M$ and $I_{sh}^M$) are set to 0.

For example, if the TIR modality (T) is missing for a sample, its feature vector $v$ and mask $mask$ would be:

$$v = \left[I_{sp}^R, I_{sp}^N, \mathbf{0}, I_{sh}^R, I_{sh}^N, \mathbf{0}\right], mask = [1, 1, 0, 1, 1, 0]. \tag{5}$$

This structured representation ($v$ and $mask$) is used consistently across all scenarios, including both modality-matched and modality-mismatched retrieval. In modality-matched scenarios (*e.g.*, RN-to-RN), both the query and gallery samples will have the same modalities present (or absent), resulting in identical structures for their masks (*e.g.*, [1, 1, 0, 1, 1, 0] for both if T is missing). In modality-mismatched scenarios (*e.g.*, R-to-N), the query and gallery samples will inherently have different available modalities. Their respective $v$ and $mask$ representations are generated according to the rules above based on their own available data. For instance, the RGB query sample would have $v_q = [I_{sp}^R, \mathbf{0}, \mathbf{0}, I_{sh}^R, \mathbf{0}, \mathbf{0}]$ and $mask_q = [1, 0, 0, 1, 0, 0]$. And the NIR gallery sample would have $v_g = [\mathbf{0}, I_{sp}^N, \mathbf{0}, \mathbf{0}, I_{sh}^N, \mathbf{0}]$ and $mask_g = [0, 1, 0, 0, 1, 0]$.

The subsequent retrieval process utilizes both the feature vectors ($v_q$, $v_g$) and their corresponding availability masks ($mask_q$, $mask_g$) to compute a similarity score that respects the decoupled nature of the features and handles missing modalities robustly. We define the total similarity, $Sim(v_q, v_g)$, as the combine of two components, *i.e.*, the modality-specific similarity $Sim_{sp}$ and the modality-shared similarity $Sim_{sh}$. The modality-specific similarity compares modality-specific features between identical modalities only. The similarity is calculated as the sum of dot products between corresponding specific features, masked for joint availability:

$$Sim_{sp} = \frac{1}{sum(mask_{sp,q}) * sum(mask_{sp,g})} \sum_{M \in \{R,N,T\}} \left((I_{sp,q}^M)^T(I_{sp,g}^M)\right) \cdot mask_{sp,q}^M \cdot mask_{sp,g}^M. \tag{6}$$

The modality-shared similarity computes the similarity between all pairs of available shared features across modalities. Define $V_{sh,q}$ be a matrix with shared features $I_{sh,q}^R$, $I_{sh,q}^N$, $I_{sh,q}^T$ (or zero vectors if unavailable) as columns, and similarly for $V_{sh,g}$, $mask_{sh,q}$ and $mask_{sh,g}$. We first compute the matrix of all pairwise shared feature similarities:

$$S_{sh} = V_{sh,q}^T V_{sh,g}. \tag{7}$$

This results in a $3 \times 3$ matrix where $(S_{sh})_{ij}$ is the dot product $(I_{sh,q}^i)^T(I_{sh,g}^j)$ for $i, j \in \{R, N, T\}$. To account for missing modalities, we define a pairwise availability mask matrix with the outer product of the shared masks:

$$M_{sh\_pair} = mask_{sh,q}^T \cdot mask_{sh,g}, \tag{8}$$

where $(M_{sh\_pair})_{ij}$ is 1 if and only if both the $i$-th shared feature of the query and the $j$-th shared feature of the gallery are present. The total shared similarity is the sum of all valid pairwise similarities, obtained by Hadamard product of $S_{sh}$ and $M_{sh\_pair}$:

$$Sim_{sh} = \frac{1}{sum(M_{sh\_pair})} \sum_{i,j} (S_{sh} \odot M_{sh\_pair})ij. \tag{9}$$

Finally, the total similarity score is:

$$Sim_{total}(v_q, v_g) = (Sim_{sp} + Sim_{sh})/2. \tag{10}$$

This approach, based on a fixed-size zero-padded feature vector and an explicit availability mask, provides a flexible and robust mechanism for handling arbitrary combinations of modality presence and absence, effectively adapting to all modality-matched and mismatched object ReID scenarios addressed by our method.

### 3.3 Modality-aware Metric Learning

After MDL, the features are separated into modality-specific and modality-shared components. To achieve more complete and thorough feature decoupling, we propose a modality-aware metric learning approach. Note that, during the training phase, we assume all the modalities for the samples are available. The core objective of this metric learning is twofold: 1) **Enhance Shared Feature Consistency**: Promote high similarity between modality-shared features irrespective of their originating modality, which is crucial to bridge the modality gap in mismatched retrieval scenarios. 2) **Reinforce Specific Feature Purity**: Ensure modality-specific features are distinct from each other and also orthogonal to all modality-shared features. This preserves unique modality characteristics and prevents leakage between specific and shared representations.

To achieve these goals simultaneously, we define a representation orthogonality loss, $L_{ROL}$ , that minimizes the discrepancy between the computed pairwise similarities of the decoupled features and a predefined target similarity structure. Within our framework, we operate on the feature vector $v$ and its availability mask $mask$ generated for each sample, as defined previously. We compute the $6 \times 6$ pairwise similarity matrix $Vsim$ based on the $L_2$-normalized vector $v$:

$$V_{sim}(i,j) = v_i^T v_j, \quad \text{for } i, j \in \{1, \ldots, 6\}. \tag{11}$$

We then define the ideal target similarity matrix as:

$$A = \begin{bmatrix} 1 & 0 & 0 & 0 & 0 & 0 \\ 0 & 1 & 0 & 0 & 0 & 0 \\ 0 & 0 & 1 & 0 & 0 & 0 \\ 0 & 0 & 0 & 1 & 1 & 1 \\ 0 & 0 & 0 & 1 & 1 & 1 \\ 0 & 0 & 0 & 1 & 1 & 1 \end{bmatrix}. \tag{12}$$

Finally, the loss $L_{ROL}$ is then formulated as the sum of squared errors between the computed similarities $V_{sim}$ and the target similarities $A$, masked by $M_{pair}$:

$$L_{ROL} = \sum_{i=1}^{6} \sum_{j=1}^{6} \left( (V_{sim}(i,j) - A(i,j))^2 \right). \tag{13}$$

Additionally, the concept of triplet loss is incorporated to further enhance the model's feature decoupling capability. In other words, the goal of feature decoupling is to achieve a synergistic effect and prevent information from collapsing into modality-specific representations. To this end, as shown in Fig. 2, we introduce a knowledge discrepancy loss (KDL) to enforce the complementarity between shared and modality-specific features, ensuring that their combination yields better retrieval performance than using either alone. Specifically, within a batch of samples, given an anchor $a$, the positive samples $P_a$ denotes the samples with the same label as $a$ while the negative samples $N_a$ denotes the opposite samples. The distances using specific feature from the anchor $a$ to the batch of positive or negative samples are computed as follows:

$$d_S(a,T) = \{\|I_S(a) - I_S(t)\|_2 \mid t \in T\},$$
$$\text{where } T \in \{P_a, N_a\}, S \in \{sp + sh, sp, sh\}, \tag{14}$$

where $S$ indicates the features used (modality-specific one, modality-shared one, or the combined one). Note that we omit the superscript representing the modality for simplicity, considering the modalities are completed, and the features of RGB, NIR, TIR modalities are concatenated to calculate the distance. Then, for the combination of shared and modality-specific features, the maximum positive sample distance should be as small as possible compared to using a single type of feature, and the minimum negative sample distance should be as large as possible compared to using a single type of feature. As shown in Fig. 2, this formulation can be described as: $\max(d_{sp+sh}(a,p)) > \max(d_{sp}(a,p)), \max(d_{sh}(a,p))$ and $\min(d_{sp+sh}(a,n)) < \min(d_{sp}(a,n)), \min(d_{sh}(a,n))$. Therefore, we define the knowledge discrepancy loss $L_{KDL}$ as:

$$L_{KDL} = \|D_p - 0\|_1 + \|D_n - 1\|_1,$$
$$\text{where } D_p = \frac{\max(d_{sp+sh}(a,p))}{\max(d_{sp+sh}(a,p)) + \max(d_{sp}(a,p)) + \max(d_{sh}(a,p))}, \tag{15}$$
$$D_n = \frac{\min(d_{sp+sh}(a,n))}{\min(d_{sp+sh}(a,n)) + \min(d_{sp}(a,n)) + \min(d_{sh}(a,n))}.$$

It is worth noting that detach operations are applied to $d_{sp}(a, n)$ and $d_{sh}^{RNT}(a, n)$ to avoid gradient computation. Thus, minimizing $D_p$ reduces the farthest positive sample distance for the combined shared and modality-specific features, while minimizing $\|D_n - 1\|_1$ increases the nearest negative sample distance. Consequently, improving the retrieval capability of the combined features enhances the performance of model. In summary, the loss function for modality-aware metric learning is defined as follows:

$$L_{MML} = w_1 \times L_{ROL} + w_2 \times L_{KDL}. \tag{16}$$

## 3.4 Objective Function

As illustrated in Fig. 2, our model incorporates two primary loss functions: one for the ViT backbone and another for modality-aware metric learning. For the ViT backbone, we adopt label smoothing cross-entropy loss $L_{ce}$ and triplet loss $L_{tri}$ following previous research to optimize the representation. In summary, the total loss function is as follows:

$$L = L_{ce} + L_{tri} + L_{MML}. \tag{17}$$

# 4 Experiments

## 4.1 Datasets and Evaluation Protocols

We evaluate the proposed **MDReID** framework on three publicly available datasets spanning both person (RGBNT201 [24]) and vehicle re-identification (RGBNT100 [8] and MSVR310 [26]) tasks. For evaluating modality mismatch scenarios, we focus on representative cross-modal configurations, including RT-to-NT, RT-to-N, R-to-N, and R-to-NT. Here, NIR is the second most widely used modality in surveillance systems after RGB, and RGB-to-NIR research continues to attract significant interest. Therefore, we evaluate our method in four R-to-N-based application scenarios. Performance is measured using mean Average Precision (mAP) and Cumulative Matching Characteristics (CMC) at ranks 1, 5, and 10 (R-1, R-5, R-10). We also report key complexity indicators, including the number of trainable parameters and floating point operations (FLOPs), to quantify the efficiency of the model.

**RGBNT201** [24] is a multi-modality person re-identification dataset designed to overcome the limitations of single-modal imaging in challenging surveillance scenarios. It was captured on a university campus using four non-overlapping camera views with a synchronized triple-camera system that simultaneously records RGB, near-infrared (NIR), and thermal-infrared (TIR) images. The dataset contains 201 identities, each represented by at least 20 non-adjacent image triplets, resulting in 4,787 images per modality. The collection covers a range of real-world conditions, including severe illumination changes, occlusion, viewpoint variations, and background clutter. The dataset is split into 141 identities for training, 30 for validation, and 30 for testing, with the entire test set serving as the gallery and 10 records per identity sampled as probes. RGBNT201 thus provides a robust benchmark for evaluating multi-modal fusion strategies and addressing missing-modality issues in person re-identification.

**RGBNT100** [8] contains 17,250 spatially aligned image triples from 100 vehicles captured under uniform conditions. Each triple includes RGB, NIR, and TIR images. Fifty vehicles (8,675 triples) are used for training, and the remaining 50 vehicles (8,575 triples) form the testing/gallery set, with 1,715 triples randomly selected as queries. The RGB and NIR images are recorded at a resolution of 1920×1080, while the TIR images are captured at 640×480, all at a consistent frame rate of 25 fps. TIR images provide thermal information that is robust to illumination changes, further enhancing the multi-spectral data for challenging vehicle re-identification scenarios.

**MSVR310** [26] is a high-quality multi-spectral vehicle re-identification benchmark captured under diverse, challenging conditions. It contains 2,087 triplet samples (6,261 images in total) from 310 vehicles. Each sample includes spatially aligned images from three modalities: RGB, NIR, and TIR. RGB images are captured by a 360 D866 camera during the day and by a Mi8 mobile phone at night, NIR images are obtained with the 360 D866 in near-infrared mode, and TIR images are recorded with a FLIR SC620 camera at 640×480 resolution. The number of samples per vehicle ranges from 2 to 20, and time labels are provided for cross-time matching. The dataset is divided into 1,032 samples from 155 vehicles for training and 1,055 samples from 155 vehicles for testing/gallery, with 591 samples

Table 1: **Performance comparison on RGBNT201, RGBNT100, and MSVR310.** The best and second results are in bold and underlined, respectively.

| | RGBNT201 | | | | | RGBNT100 | | MSVR310 | |
| Method | mAP | R-1 | R-5 | R-10 | Method | mAP | R-1 | mAP | R-1 |
|---|---|---|---|---|---|---|---|---|---|
| **Single** | | | | | | | | | |
| MUDeep [39] | 23.8 | 19.7 | 33.1 | 44.3 | DMML [40] | 58.5 | 82.0 | 19.1 | 31.1 |
| HACNN [41] | 21.3 | 19.0 | 34.1 | 42.8 | BoT [42] | 78.0 | 95.1 | 23.5 | 38.4 |
| MLFN [43] | 26.1 | 24.2 | 35.9 | 44.1 | Circle Loss [44] | 59.4 | 81.7 | 22.7 | 34.2 |
| PCB [45] | 32.8 | 28.1 | 37.4 | 46.9 | HRCN [46] | 67.1 | 91.8 | 23.4 | 44.2 |
| OSNet [47] | 25.4 | 22.3 | 35.1 | 44.7 | TransReID [1] | 75.6 | 92.9 | 18.4 | 29.6 |
| CAL [48] | 27.6 | 24.3 | 36.5 | 45.7 | AGW [16] | 73.1 | 92.7 | 28.9 | 46.9 |
| **Multi** | | | | | | | | | |
| HAMNet [8] | 27.7 | 26.3 | 41.5 | 51.7 | GAFNet [25] | 74.4 | 93.4 | - | - |
| PFNet [24] | 38.5 | 38.9 | 52.0 | 58.4 | GraFT [49] | 76.6 | 94.3 | - | - |
| IEEE [9] | 47.5 | 44.4 | 57.1 | 63.6 | GPFNet [28] | 75.0 | 94.5 | - | - |
| DENet [27] | 42.4 | 42.2 | 55.3 | 64.5 | PHT [35] | 79.9 | 92.7 | - | - |
| UniCat [36] | 57.0 | 55.7 | - | - | UniCat [36] | 79.4 | 96.2 | - | - |
| HTT [10] | 71.1 | 73.4 | 83.1 | 87.3 | CCNet [26] | 77.2 | 96.3 | 36.4 | 55.2 |
| EDITOR [11] | 66.5 | 68.3 | 81.1 | 88.2 | EDITOR [11] | 82.1 | 96.4 | 39.0 | 49.3 |
| RSCNet [31] | 68.2 | 72.5 | - | - | RSCNet [31] | 82.3 | **96.6** | 39.5 | 49.6 |
| TOP-ReID [12] | 72.3 | 76.6 | 84.7 | 89.4 | TOP-ReID [12] | 81.2 | 96.4 | 35.9 | 44.6 |
| **MDReID (Ours)** | **82.1** | **85.2** | **90.3** | **92.6** | **MDReID (Ours)** | **85.3** | 95.6 | **51.0** | **68.9** |

(from 52 vehicles) used as queries. MSVR310 provides a comprehensive platform for addressing intra-class appearance variations and modality differences to support robust vehicle re-identification.

## 4.2 Implementation Details

All experiments are implemented in PyTorch and conducted on a single NVIDIA RTX 4090 GPU with CUDA 12.5 and Python 3.8. We adopt the CLIP-Base [37] visual encoder as the backbone. Input images are resized to $256 \times 128$ for RGBNT201 and $128 \times 256$ for RGBNT100 and MSVR310. We employ random horizontal flipping, cropping, and erasing [38] for data augmentation. The model is optimized using Adam with a batch size of 64. The base learning rate is initialized at $3.5 \times 10^{-4}$, while the visual encoder is fine-tuned with a reduced rate of $5 \times 10^{-6}$. Training is performed for 50 epochs.

## 4.3 Comparison with State-of-the-Art Methods

**Multi-spectral Object ReID (RNT-to-RNT)** We compare our MDReID against a range of state-of-the-art single-spectral and multi-spectral approaches on RGBNT201, RGBNT100, and MSVR310. For person ReID on RGBNT201, as shown in Table 1, the TOP-ReID [12] achieves 72.3% mAP, 76.6% R-1, 84.7% R-5, and 89.4% R-10. In comparison, our MDReID outperforms these results by 9.8%, 8.6%, 5.6%, and 3.2%, respectively, highlighting the effectiveness of our proposed MDReID. For vehicle re-ID, MDReID achieves the highest mAP on RGBNT100 (85.3%) and surpasses the next-best approach on MSVR310 by a significant margin of 11.5% in mAP and 13.7% in R-1. These results clearly demonstrate the superior discriminative power and robustness of our modality-decoupled design in multi-spectral re-ID scenarios where all modalities are available.

**Evaluation on Missing-spectral Scenarios.** In practical applications, sensor limitations or environmental factors often lead to incomplete modality inputs. Following the setting of TOP-ReID [12], we evaluate the robustness of our method under all six possible missing-modality configurations across the RGB, NIR, and TIR channels using the RGBNT201 dataset. As shown in Table 2, our MDReID consistently outperforms the TOP-ReID, achieving an average improvement of 10.0% in mAP and 9.4% in Rank-1 accuracy across all missing-modality scenarios. These results demonstrate that our modality-decoupled framework remains highly effective even when partial modality information is absent, highlighting its strong generalization ability in incomplete multispectral settings.

**Evaluation on Modality-mismatched Scenarios.** To evaluate the generalizability of our approach under realistic deployment conditions, we benchmark MDReID across four representative modality-mismatched scenarios on three public datasets. We compare against two strong baselines, TOP-ReID [12] and EDITOR [11], by reproducing their results using official open-source implementations. As shown in Table 3, EDITOR is designed exclusively for modality-matched settings, which exhibits limited effectiveness when modalities differ across query and gallery. Although TOP-ReID attempts

Table 2: **Performance of missing-modality settings on RGBNT201.** "M (X)" means missing the X image modality. The best and second results are in bold and underlined, respectively.

| | Methods | M (RGB) | | M (NIR) | | M (TIR) | | M (RGB+NIR) | | M (RGB+TIR) | | M (NIR+TIR) | | Average | |
|---|---|---|---|---|---|---|---|---|---|---|---|---|---|---|---|
| | | *m*AP | R-1 | *m*AP | R-1 | *m*AP | R-1 | *m*AP | R-1 | *m*AP | R-1 | *m*AP | R-1 | *m*AP | R-1 |
| Single | MUDeep [39] | 19.2 | 16.4 | 20.0 | 17.2 | 18.4 | 14.2 | 13.7 | 11.8 | 11.5 | 6.5 | 12.7 | 8.5 | 15.9 | 12.9 |
| | HACNN [41] | 12.5 | 11.1 | 20.5 | 19.4 | 16.7 | 13.3 | 9.2 | 6.2 | 6.3 | 2.2 | 14.8 | 12.0 | 13.3 | 10.7 |
| | MLFN [43] | 20.2 | 18.9 | 21.1 | 19.7 | 17.6 | 11.1 | 13.2 | 12.1 | 8.3 | 3.5 | 13.1 | 9.1 | 15.6 | 12.4 |
| | PCB [45] | 23.6 | 24.2 | 24.4 | 25.1 | 19.9 | 14.7 | 20.6 | 23.6 | 11.0 | 6.8 | 18.6 | 14.4 | 19.7 | 18.1 |
| | OSNet [47] | 19.8 | 17.3 | 21.0 | 19.0 | 18.7 | 14.6 | 12.3 | 10.9 | 9.4 | 5.4 | 13.0 | 10.2 | 15.7 | 12.9 |
| Multi | PFNet [24] | - | - | 31.9 | 29.8 | 25.5 | 25.8 | - | - | - | - | 26.4 | 23.4 | - | - |
| | DENet [27] | - | - | 35.4 | 36.8 | 33.0 | 35.4 | - | - | - | - | 32.4 | 29.2 | - | - |
| | TOP-ReID [12] | 54.4 | 57.5 | 64.3 | 67.6 | 51.9 | 54.5 | 35.3 | 35.4 | 26.2 | 26.0 | 34.1 | 31.7 | 44.4 | 45.4 |
| | **MDReID (Ours)** | **67.0** | **67.9** | **75.8** | **80.9** | **61.7** | **59.8** | **48.6** | **51.1** | **31.4** | **29.2** | **42.1** | **40.0** | **54.4** | **54.8** |

Table 3: **Performance of modality-mismatched Scenarios.** We show the best average score in bold.

| Methods | | RT-to-NT | | RT-to-N | | R-to-N | | R-to-NT | | Average | |
|---|---|---|---|---|---|---|---|---|---|---|---|---|
| | | *m*AP | R-1 | *m*AP | R-1 | *m*AP | R-1 | *m*AP | R-1 | *m*AP | R-1 |
| RGBNT201 | EDITOR [11] | 27.3 | 27.9 | 2.80 | 0.0 | 4.0 | 2.3 | 4.3 | 4.1 | 8.5 | 7.5 |
| | TOP-ReID [12] | 43.0 | 44.6 | 14.4 | 13.3 | 15.4 | **14.0** | 11.9 | 8.7 | 18.2 | 18.0 |
| | **MDReID (Ours)** | **53.1** | **51.7** | **16.7** | **13.8** | **16.6** | 11.1 | **15.1** | **10.9** | **21.6** | **19.1** |
| RGBNT100 | EDITOR [11] | 42.1 | 59.9 | 2.6 | 0.8 | 2.8 | 1.5 | 2.7 | 1.5 | 11.9 | 15.5 |
| | TOP-ReID [12] | 59.0 | 81.8 | 21.9 | 28.5 | 26.2 | 34.0 | 21.7 | 25.4 | 26.8 | 36.1 |
| | **MDReID (Ours)** | **69.4** | **85.5** | **39.6** | **47.9** | **45.4** | **56.4** | **37.6** | **41.8** | **38.6** | **47.4** |
| MSVR310 | EDITOR [11] | 6.4 | 11.0 | 2.2 | 2.5 | 1.6 | 0.2 | 1.7 | 1.5 | 2.5 | 3.4 |
| | TOP-ReID [12] | 18.4 | 30.5 | 12.9 | 19.5 | 13.7 | 21.3 | 12.6 | 18.8 | 11.2 | 17.8 |
| | **MDReID (Ours)** | **35.1** | **52.5** | **24.7** | **34.5** | **28.6** | **39.8** | **27.0** | **36.0** | **22.1** | **31.7** |

to address missing modalities via reconstruction, its performance remains constrained due to the ill-posed nature of the reconstruction problem. In contrast, our MDReID framework consistently improves the average mAP by 3.4%, 11.8%, and 10.9% on the RGBNT201, RGBNT100, and MSVR310 datasets, respectively. In addition, we train four expert models on the RGBNT201 dataset for four specific scenarios. Results show that our method outperforms these experts by 9.2% on average in mAP and by 8.9% in R-1. Importantly, our single-model approach adapts to all four scenarios and offers greater flexibility. In summary, these results highlight the robustness and adaptability of MDReID in handling modality-mismatched ReID, affirming its effectiveness capable of supporting arbitrary modality combinations in both query and gallery inputs.

## 4.4 Ablation Study

To evaluate the effectiveness of each component in our proposed MDReID framework, we conduct comprehensive ablation studies on the RGBNT201 dataset. To ensure a thorough evaluation across both modality-matched and modality-mismatched scenarios, we report the average performance over 8 evaluation settings, including RNT-to-RNT, RT-to-RT, RT-to-NT, RT-to-N, R-to-N, and R-to-NT. The detailed results are presented in Table 4 (a). Note that configuration "1" corresponds to using a single classifier for all modalities, while configuration "3" corresponds to assigning a separate classifier to each modality. Ablation results show that, compared with one classifier, modality-specific classifiers effectively improve performance, increasing mean mAP by 13.4% and mean R-1 by 13.7%.

**Modality Decoupled Learning** Our base framework integrates a shared vision transformer, utilizing label smoothing cross-entropy along with triplet loss for optimization. To better address modality-mismatched object re-identification, we design a modality decoupling learning (MDL) that separates features into shared and specific representations. Experimental results demonstrate that this module boosts performance, with an average increase of 11.5% in mAP and 11.1% in Rank-1. This demonstrates that modality decoupling not only benefits re-identification in mismatched settings but also enhances retrieval performance in matched scenarios.

**Modality-aware Metric Learning** To further decouple shared and specific features, we design a modality-aware metric learning (MML) loss function. This loss consists of two components: the first aligns shared features across multiple modalities, while the second enhances the decoupling of features based on the aligned representation. Experimental results show that introducing ROL improves average mAP and Rank-1 by 1.8% and 2.7%, respectively. Similarly, introducing KDL increases mAP by 0.5% and Rank-1 by 1.7%. When both components are combined, the average mAP and Rank-1 improve by 3.8% and 4.1%, respectively. These results indicate that ROL significantly

Table 4: **Ablation study and hyper-parameter settings of MDReID.** We show the best average score in bold.

(a) **Ablation study.** MDL, $L_{ROL}$, and $L_{KDL}$ indicate the Modality Decoupled Learning, Representation Orthogonality Loss (ROL), and Knowledge Discrepancy Loss (KDL), respectively.

| Index | MDL | $L_{ROL}$ | $L_{KDL}$ | $m$AP | R-1 |
|---|---|---|---|---|---|
| 1 | ✗ | ✗ | ✗ | 27.8 | 27.1 |
| 2 | ✓ | ✗ | ✗ | 39.4 | 38.2 |
| 3 | ✓ | ✓ | ✗ | 41.2 | 40.8 |
| 4 | ✓ | ✗ | ✓ | 39.9 | 40.9 |
| 5 | ✓ | ✓ | ✓ | **43.2** | **42.3** |

(b) **Performance under different $w_1$.** The optimal performance achieves when $w_1$ is set to 1.5.

| $w_1$ | $m$AP | R-1 |
|---|---|---|
| 0.5 | 39.6 | 38.1 |
| 1.0 | 40.3 | 40.0 |
| 1.5 | **41.2** | **40.8** |
| 2.0 | 38.9 | 38.1 |
| 3.0 | 38.2 | 37.0 |

(c) **Performance under different $w_2$.** The optimal performance achieves when $w_2$ is set to 5.25.

| $w_2$ | $m$AP | R-1 |
|---|---|---|
| 4.5 | 41.7 | 41.1 |
| 5.0 | 40.2 | 39.6 |
| 5.25 | **43.2** | **42.3** |
| 5.5 | 41.6 | 41.3 |
| 6.0 | 41.9 | 40.4 |

enhances re-identification accuracy in mismatch scenarios. Moreover, while KDL alone provides limited improvement, its combination with ROL effectively boosts overall model performance.

**Modality Decoupled Learning** We design the KDL objective to enforce a knowledge gap between specific and shared features. Without KDL, the model may degrade into relying only on shared features. Therefore, we ensure that combining specific and shared features yields higher retrieval performance than using either feature alone. To achieve this, we apply formulation 15 to impose distance constraints in the feature space based on the described relations. As shown in Table 4 (a), index 3 and 5, adding KDL improves mAP by 2.0% and Rank-1 by 1.5% across multiple scenarios. In summary, the MDReID framework, comprising the modality decoupled learning and modality-aware metric learning, achieves an average performance of 43.2% mAP, 42.3% Rank-1, 50.2% Rank-5, and 54.9% Rank-10 across various scenarios, including both modality-matched and modality-mismatched cases. Its modular design effectively decouples shared and specific features, enhancing object re-identification accuracy under multi-spectral conditions while maintaining robust adaptability across diverse modalities.

## 4.5   Discussions[4]

**Hyper-parameter settings of MDReID** In Eq. (16), we introduce two hyperparameters $w_1$ and $w_2$ to balance the importance of ROL and KDL. Therefore, this part evaluates the performance under different hyperparameter settings and shows the results in Table 4. We first vary $w_1$ without KDL, observing that the model achieves optimal performance at $w_1 = 1.5$, highlighting the importance of enforcing orthogonality between shared and specific components. Subsequently, with $w_1 = 1.5$, we tune $w_2$ and find that the best results are obtained at $w_2 = 5.25$, confirming the benefit of encouraging representational synergy and complementarity. These observations validate that a contribution from both ROL and KDL is crucial for maximizing retrieval accuracy.

## 5   Conclusion

In this paper, we propose MDReID, an image-level any-to-any object re-identification framework that supports retrieval across arbitrary query-gallery modality combinations. Our approach leverages Modality-Decoupled Learning (MDL) to explicitly disentangle modality-shared and modality-specific features, while Modality-aware Metric Learning (MML) refines the feature space to enhance discrimination. Extensive evaluations on the RGBNT201, RGBNT100, and MSVR310 datasets validate MDReID's superior performance in both modality-matched and modality-mismatched scenarios, demonstrating its practical potential for any-to-any ReID tasks.

**Acknowledgements.** This research was supported in part by the China Postdoctoral Science Foundation under Grant Number 2025M771584 and 2025T180439, and the National Natural Science Foundation of China under Grant 6250071985

---

[4]A comprehensive analysis of the model's computational complexity (Sec. A.1), along with additional visualization results (Sec. A.4), is provided in the supplementary material.

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

# A    Supplemental Material

## A.1    Analysis of model computational complexity

Table 5 compares the computational cost of our method with recent state-of-the-art approaches. MDReID achieves the most efficient design, requiring only **57.2M** parameters and **22.3G** FLOPs, which is significantly lower than HTT (85.6M / 33.1G), EDITOR (117.5M / 38.6G), and TOP-ReID (278.2M / 34.5G). Despite its efficient architecture, MDReID consistently outperforms these heavier models in accuracy, demonstrating a superior trade-off between performance and efficiency. This highlights the effectiveness of our MDL and MML designs, which provide robust alignment in all kinds of query-gallery combinations without incurring large computational overhead. Furthermore, we evaluate the total feature generation and retrieval times of different models on the RGBNT201 test set under the RNT-to-RNT, RT-to-NT, and R-to-N scenarios. Theoretically, our method reduces retrieval cost compared to Top-ReID. For example, in the RT-to-NT scenario, we only perform T-to-T (specific features) and R-to-T (shared features) matching, whereas Top-ReID first generates missing-modality features before conducting a full RNT-to-RNT matching. Experimental results confirm this, showing that our method incurs no additional time and generates features faster.

Table 5: **Comparison of computational cost with recent methods.** We show the best result in bold.

| Methods | Params(M) | Flops(G) | RNT-to-RNT(s) | | RT-to-NT(s) | | R-to-N(s) | |
|---|---|---|---|---|---|---|---|---|
| | | | Extract | Retrieval | Extract | Retrieval | Extract | Retrieval |
| HTT [10] | 85.6 | 33.1 | 8.32 | **0.12** | - | - | - | - |
| EDITOR [11] | 117.5 | 38.6 | 31.12 | **0.12** | 30.99 | 0.12 | 30.98 | **0.12** |
| TOP-ReID [12] | 278.2 | 34.5 | 71.78 | **0.12** | 73.32 | **0.11** | 73.71 | **0.12** |
| MDReID (Ours) | **57.2** | **22.3** | **3.27** | 0.14 | **2.90** | **0.11** | **3.14** | **0.12** |

## A.2    Analysis of specific and shared features

In MDReID, both shared and modality-specific features are essential. For the RT-to-NT task, using only shared features overlooks the unique characteristics of the T modality, while using only modality-specific features prevents valid R-to-N matching. To quantify this, we evaluate RT-to-NT performance under three settings: shared features only, modality-specific features only, and both combined. As shown in Table 6, combining shared and modality-specific features yields the best performance.

Table 6: **Performance analysis under different features.** We show the best result in bold.

| Feature | *m*AP | R-1 |
|---|---|---|
| Specific feature | 52.0 | 50.5 |
| Shared feature | 52.7 | 51.2 |
| Specific and shared feature | **53.1** | **51.7** |

## A.3    Comparison with cross-modality method

We compare our method with the cross-modality object re-identification model DEEN [50] on the RGBNT201 dataset under the R-to-N scenario. Table 7 shows that even when DEEN is specially trained for R-to-N, its mean precision is 3.5% lower than ours. Notably, our model demonstrates stronger generalization across diverse application scenarios.

Table 7: **Comparison with cross-modality method.** We show the best result in bold.

| Feature | *m*AP | R-1 | Average |
|---|---|---|---|
| DEEN [50] | 11.5 | 9.2 | 10.35 |
| Ours | 16.6 | 11.1 | 13.85 |

## A.4    Visualization

To better illustrate the impact of different components on feature disentanglement, we visualize the learned modality-specific and modality-shared features across RGB, NIR, and TIR using t-SNE, as shown in Figure 3. Notably, the baseline method does not use MDL, so it does not separate specific and shared features and only outputs fused features, hence no corresponding visualization.

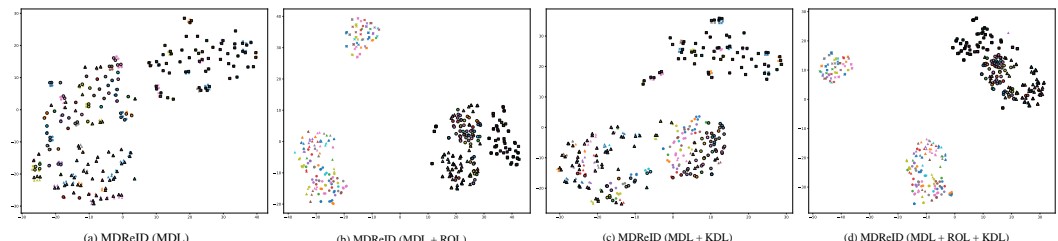

(a) MDReID (MDL)     (b) MDReID (MDL + ROL)     (c) MDReID (MDL + KDL)     (d) MDReID (MDL + ROL + KDL)

Figure 3: The visualization of the ablation study. Triangles, circles, and X symbols correspond to the RGB, NIR, and TIR modalities, respectively. Black borders highlight shared features for clarity. Without ROL, distinguishing between shared and modality-specific features is challenging, but its introduction effectively clusters the shared features.

Specifically, in (a), with only MDL applied, the features exhibit noticeable overlap, and the separation between modality-specific and shared components remains unclear. In (c), the addition of KDL slightly improves the clustering and separation of shared and specific features. In contrast, (b) shows that introducing ROL significantly enhances feature orthogonality, forming clearer boundaries between shared and modality-specific clusters. Finally, (d) demonstrates that the combination of both ROL and KDL yields the most structured and disentangled feature space, with shared features tightly clustered and well-separated from their modality-specific counterparts. These results confirm the complementary roles of ROL and KDL in refining the representation space.

## A.5 Limitations and Broader Impact

Although our work is the first to explore object re-identification under the uncertain query-gallery modality setting and achieves significant performance improvements, the current accuracy is still insufficient for deployment in real-world applications. Furthermore, existing datasets are limited in both scale and modality diversity, making it unclear how well MDReID would generalize to larger and more complex datasets involving a broader range of modality types. Nevertheless, MDReID represents the first effort to address the *any-to-any* object re-identification task, marking a significant step toward this emerging direction. We believe our work will inspire new perspectives and challenges in the re-identification community.

