# OpenReview forum: "MDReID: Modality-Decoupled Learning for Any-to-Any Multi-Modal Object Re-Identification"
_NeurIPS.cc/2025/Conference — NeurIPS 2025 spotlight_

### Official Review · Reviewer_dspH · 2025-06-27

**Clarity:** 4
**Significance:** 3
**Originality:** 3
**Rating:** 6
**Confidence:** 4

**Summary:**

This paper proposes MDReID, a framework for object re-identification across that allows any query and gallery combinations from RGB, NIR, and TIR images. Unlike prior work that focuses on fixed modality pairs, MDReID handles arbitrary modality combinations through two key components: Modality-Decoupled Learning (MDL), which separates features into modality-shared and modality-specific parts, and Modality-aware Metric Learning (MML), which ensures these parts are complementary via orthogonality and discrepancy constraints. This design enables robust matching under varying or missing modalities. Extensive experiments on RGBNT201, RGBNT100, and MSVR310 show that MDReID achieves state-of-the-art performance with significantly fewer parameters and lower computation cost.

**Questions:**

1. This paper appears to be the only one among all compared methods that adopts CLIP as the backbone. It would be helpful to clarify whether this choice contributes additional performance gains.

2. The discussion on KDL may be somewhat insufficient. While the authors have dedicated adequate space to describing how KDL is implemented, they have not clearly articulated how it contributes to the model’s performance or why it is designed in its current form. I encourage the authors to elaborate on the underlying design rationale of KDL in their rebuttal.

3. I believe that using a classifier to explicitly distinguish between different modalities is a simple yet effective strategy. However, it appears that the authors did not adopt this approach in the current work. I would suggest including a comparison to assess whether incorporating such a strategy could lead to further performance improvements.

4. In Table 2, the performance of TOP-ReID appears to be mistakenly highlighted in bold.

**Ethical Concerns:**

["NO or VERY MINOR ethics concerns only"]

**Final Justification:**

This paper proposes MDReID, a novel framework for object re-identification. It handles arbitrary modality combinations through two key components: Modality-Decoupled Learning (MDL) and Modality-aware Metric Learning (MML). This design enables robust matching under varying or missing modalities.

**Limitations:**

Yes, this work has addressed their limitations in ‘Supplemental Material’.

**Paper Formatting Concerns:**

The format of the article meets the requirements of NeurIPS 2025.

**Quality:**

3

**Strengths And Weaknesses:**

The strengths of this paper are quite clear. The proposed any-to-any object re-identification task is both novel and compelling. Existing works typically split modality-aligned and cross-modal retrieval into two separate problems. I do not believe this is a desirable direction, as it fragments the solution space and limits generalization. In contrast, this paper is the first to support arbitrary query-gallery modality combinations within a unified framework, which I consider a significant contribution.

On the downside, the overall methodology appears relatively simple. While this simplicity may be an advantage from an application standpoint, the idea of feature decoupling itself is not particularly novel within the ReID literature.

Nevertheless, I believe the paper’s contribution outweighs this concern. Its exploration of the any-to-any setting is timely and has the potential to make a notable impact on the re-identification community.

---

> ### Author Rebuttal · Authors · 2025-07-31
>
> > **Q1:** This paper appears to be the only one among all compared methods that adopts CLIP as the backbone. It would be helpful to clarify whether this choice contributes additional performance gains.
> >
>
> **A1:**  Thank you for your suggestion. To validate that CLIP contributes additional performance gains, we conduct experiments on our server to compare Top-ReID with our method, both using a CLIP-based ViT backbone. As shown in Table A, our method outperforms Top-ReID by approximately 11.1 % in mAP and 11.3 % in R-1. These results confirm the superior performance of our approach under identical settings and with the same backbone.
>
> Table A: Performance comparison on CLIP-based ViT.
>
> | Methods | mAP | R-1 |
> | --- | --- | --- |
> | TOP-ReID | 71.0 | 73.9 |
> | Ours | **82.1** | **85.2** |
>
> > **Q2:** The discussion on KDL may be somewhat insufficient. While the authors have dedicated adequate space to describing how KDL is implemented, they have not clearly articulated how it contributes to the model’s performance or why it is designed in its current form. I encourage the authors to elaborate on the underlying design rationale of KDL in their rebuttal.
> >
>
> **A2:**  Thank you for pointing out this issue. We design the KDL objective to enforce a knowledge gap between specific and shared features. Without KDL, the model may collapse into relying only on shared features. We therefore ensure that combining specific and shared features yields higher retrieval performance than using either feature alone. To achieve this, we apply the formulation in lines 222–225, which imposes distance constraints in the feature space. Table 4 validates our approach: as shown in Table A, adding KDL improves mAP by 2.0% and Rank-1 by 1.5% across multiple scenarios. In subsequent versions, we provide more detailed explanations. We have also revised this section accordingly to make it clearer.
>
> Table B: Ablation study.
>
> | Methods | mAP | R-1 |
> | --- | --- | --- |
> | Without KDL | 41.2 | 40.8 |
> | KDL | **43.2** | **42.3** |
>
> > **Q3:** I believe that using a classifier to explicitly distinguish between different modalities is a simple yet effective strategy. However, it appears that the authors did not adopt this approach in the current work. I would suggest including a comparison to assess whether incorporating such a strategy could lead to further performance improvements.
> >
>
> **A3:**  Thank you for your suggestion. Actually, in our ablation study, we perform relevant experiments, but our inadequate description of the baseline may lead to misunderstanding. Specifically, in Table 4(a), configuration “1” corresponds to using a single classifier for all modalities, while configuration “3” corresponds to assigning a separate classifier to each modality. The ablation results show that modality-specific classifiers effectively improve performance, boosting mean mAP by 13.4% and mean R-1 by 13.7%.
>
> Table C: Comparison of different classifiers.
>
> | Methods | mAP | R-1 |
> | --- | --- | --- |
> | One classifier  | 27.8 | 27.1 |
> | different modalities classifier | **41.2** | **40.8** |
>
> > **Q4:** In Table 2, the performance of TOP-ReID appears to be mistakenly highlighted in bold.
> >
>
> **A4:**  Thank you for your suggestion. We will make the corresponding changes in future versions.

---

### Official Review · Reviewer_oBaa · 2025-06-28

**Clarity:** 4
**Significance:** 4
**Originality:** 4
**Rating:** 5
**Confidence:** 5

**Summary:**

This paper introduces MDReID, a general-purpose object Re-ID framework designed for modality-mismatched and modality-matched scenarios. It addresses the practical limitation of existing methods that assume consistent modalities. MDReID consists of two key modules: MDL decomposes features into modality-shared and modality-specific components. MML enhances feature discrimination via orthogonality and complementarity. The method is validated on three multi-modality Re-ID benchmarks and shows significant improvements over baselines in both modality-matched and mismatched settings.

**Questions:**

See Weakness.

**Ethical Concerns:**

["NO or VERY MINOR ethics concerns only"]

**Final Justification:**

The response addresses my concerns.

**Limitations:**

Yes

**Paper Formatting Concerns:**

No formatting issues were found. The paper adheres to the NeurIPS 2025 formatting guidelines.

**Quality:**

4

**Strengths And Weaknesses:**

Strengths:
This paper clearly identifies the modality inconsistency issue in real-world ReID systems and proposes a targeted solution.
This paper Introduces a dual mechanism of modality decoupling (MDL) and metric learning (MML), with a well-structured design.
The performance improvements are validated across multiple benchmark datasets, supported by strong empirical evidence.
Weaknesses:
1. Lack of a Clear Classification Framework in Related Work: The related work lacks a clear technical taxonomy. The current version lists existing methods but does not organize them by core challenges (e.g., Conventional multi-modal methods,ViT-based, missing-modality methods). It also lacks analysis of each method's key ideas and limitations.
2. Redundant or Delayed Definition of Technical Terms (e.g., ViT): Several technical terms (e.g., ViT) are either repeated or introduced only after they have already been used multiple times.
3. Binary Availability Mask Explanation is Overly Verbose: The explanation of the binary availability mask spans over 0.5 pages. The logic of zero-padding can be simplified or moved to the supplementary material. The current version distracts from the main technical innovation.
4. Knowledge Discrepancy Loss Definition (Line 224) is Unclear: The structure of the knowledge discrepancy loss is hard to interpret.
Anyway, this is a good paper with a new task and reasonable solution, although there are some minor flaws in the details.

---

> ### Author Rebuttal · Authors · 2025-07-31
>
> > **Q1:** Lack of a Clear Classification Framework in Related Work: The related work lacks a clear technical taxonomy. The current version lists existing methods but does not organize them by core challenges (e.g., Conventional multi-modal methods,ViT-based, missing-modality methods). It also lacks analysis of each method's key ideas and limitations.
> >
>
> **A1:** Thank you for the valuable feedback. We agree that the introduction and related work section would benefit from a clearer technical taxonomy and deeper analysis. We have revised these sections following your suggestions to categorize existing methods by core challenges and include a discussion of their key ideas and limitations.
>
> > **Q2:** Redundant or Delayed Definition of Technical Terms (e.g., ViT): Several technical terms (e.g., ViT) are either repeated or introduced only after they have already been used multiple times.
> >
>
> **A2:** Thank you for the comments. We have revised the manuscript to ensure that all technical terms are clearly defined and used.
>
> > **Q3:** Binary Availability Mask Explanation is Overly Verbose: The explanation of the binary availability mask spans over 0.5 pages. The logic of zero-padding can be simplified or moved to the supplementary material. The current version distracts from the main technical innovation..
> >
>
> **A3:**  Thank you for the suggestion. We have accordingly reduced the content of the related section following your suggestion.
>
> > **Q4:** Knowledge Discrepancy Loss Definition (Line 224) is Unclear: The structure of the knowledge discrepancy loss is hard to interpret. Anyway, this is a good paper with a new task and reasonable solution, although there are some minor flaws in the details.
> >
>
> **A4:**  Thank you for pointing out this issue. We design the KDL objective to enforce a knowledge gap between specific and shared features. Without KDL, the model may degrade into relying only on shared features. We therefore ensure that combining specific and shared features yields higher retrieval performance than using either feature alone. To achieve this, we apply the formulation in lines 222–225, which imposes distance constraints in the feature space based on the mentioned relation. Table 4 validates our approach: as shown in Table A, adding KDL improves mAP by 2.0% and Rank-1 by 1.5% across multiple scenarios. We have also revised this section accordingly to make it clearer.
>
> Table A: Ablation study.
>
> | Methods | mAP | R-1 |
> | --- | --- | --- |
> | Without KDL | 41.2 | 40.8 |
> | KDL | 43.2 | 42.3 |

---

> > ### Comment · Reviewer_oBaa · 2025-08-03
> >
> > The response addresses my concerns.

---

### Official Review · Reviewer_FcJV · 2025-06-30

**Clarity:** 3
**Significance:** 3
**Originality:** 4
**Rating:** 5
**Confidence:** 5

**Summary:**

MDReID is a novel framework for multi-modal object re-identification that supports any-to-any retrieval across both modality-matched and modality-mismatched scenarios. To tackle the challenges of modality discrepancy, MDReID introduces a Modality-Decoupled Learning (MDL) strategy to explicitly separate modality-shared and modality-specific representations. In addition, a Modality-aware Metric Learning (MML) strategy is proposed, which enhances this disentanglement through a representation orthogonality loss and a knowledge discrepancy loss, encouraging the two components to capture distinct yet complementary information. The framework is comprehensively evaluated under both modality-aligned and modality-misaligned settings to demonstrate its effectiveness.

**Questions:**

1.	The any-to-any setting is highly related to both modality-matched object re-identification and cross-modality object re-identification. However, this work only considers the modality-matched side. Therefore, comparison and discussion with cross-modality methods will also be useful.
2.	During the retrieval phase. From the paper, the author considers the shared representations equally, whether the shared feature from the same modality. From the visualization results, the shared feature.
3.	Figure 3 lacks the visualization result of the baseline method. It will make audience hard to get the improvement of MDL. The same issue applies to Table 5.
4.	It would be helpful to clarify how the 4 testing scenarios under modality-mismatched conditions were selected.

**Ethical Concerns:**

["NO or VERY MINOR ethics concerns only"]

**Final Justification:**

The authors have addressed all my concerns with quantitative evidence and clear explanations.

**Limitations:**

Yes

**Paper Formatting Concerns:**

No formatting concerns.

**Quality:**

3

**Strengths And Weaknesses:**

Pros:
-	This paper introduces the novel task of any-to-any multi-modal object re-identification, which supports flexible retrieval under arbitrary query-gallery combinations across three different spectral modalities.
-	Compared to the previous works, the proposed method achieves a significant improvement in all the testing protocols.
Cons:
-	The any-to-any setting is also highly related to the cross-modality re-identification task. However, this paper may ignore this part.
-	Some details should be further explained. Some important details require further clarification. For example, what is the computational cost of the baseline method? Additionally, how the 4 testing scenarios under modality-mismatched conditions were selected?

---

> ### Author Rebuttal · Authors · 2025-07-31
>
> > **Q1:** The any-to-any setting is highly related to both modality-matched object re-identification and cross-modality object re-identification. However, this work only considers the modality-matched side. Therefore, comparison and discussion with cross-modality methods will also be useful.
> >
>
> **A1:**  Thank you for your feedback, and we incorporated additional discussions on cross-modality object re-identification. In this experiment, we evaluate the DEEN [1] algorithm on the RGBNT201 dataset under the R-to-N scenario. As shown in Table A, our results demonstrate that even with specialized training for R-to-N, DEEN achieves an average precision 3.5% lower than our method. Notably, our model exhibits stronger generalization capability across diverse application scenarios.
>
> Table A: Comparison of DEEN and Our Method in R-to-N Scenario.
>
> | R→N | mAP | R-1 | Avg |
> | --- | --- | --- | --- |
> | DEEN  | 11.5 | 9.2 | 10.35 |
> | Ours | **16.6** | **11.1** | 13.85 |
>
> [1] Diverse embedding expansion network and low-light cross-modality benchmark for visible-infrared person re-identification. CVPR 2023.
>
> > **Q2:** During the retrieval phase. From the paper, the author considers the shared representations equally, whether the shared feature from the same modality. From the visualization results, the shared feature.
> >
>
> **A2:** Thanks for your valuable suggestion. We have evaluated the performance under different weight combinations of specific features (w_sp) and shared features (w_sh), and showed the result below. The experimental results indicate that adjusting the weights does indeed lead to a slight improvement. Specifically, assigning a higher weight to specific features benefits retrieval in modality-matched scenarios, while a higher weight to shared features helps in modality-mismatched retrieval. We will include a detailed discussion on parameter settings in a future version to update the relevant experimental sections.
>
> Table B: Analysis of different weights.
>
> | (w_sh,w_sp) | RNT→RNT |  | RT→NT |  | RT→N |  | Avg |  |
> | --- | --- | --- | --- | --- | --- | --- | --- | --- |
> |  | mAP | R-1 | mAP | R-1 | mAP | R-1 | mAP | R-1 |
> | ours | 82.1 | 85.2 | 53.1 | 51.7 | 16.7 | 13.8 | 50.63 | 50.23 |
> | 1.0,0.0 | 79.6 | 83.5 | 51.9 | 51.0 | 16.6 | 11.1 | 49.37 | 48.53 |
> | 0.9,0.1 | 80.1 | 83.7 | 52.1 | 51.0 | 17.5 | 12.9 | 49.90 | 49.20 |
> | 0.8,0.2 | 80.6 | 84.1 | 52.4 | 51.0 | 18.2 | 14.5 | 50.40 | 49.87 |
> | 0.7,0.3 | 81.1 | 84.2 | 52.7 | 50.8 | **18.5** | **15.9** | 50.77 | 50.30 |
> | 0.6,0.4 | 81.6 | 84.9 | 52.9 | 51.2 | 18.0 | 14.8 | **50.83** | **50.30** |
> | 0.5,05 | 82.1 | 85.2 | 53.1 | 51.7 | 16.7 | 13.8 | 50.63 | 50.23 |
> | 0.4,0.6 | 82.6 | 85.0 | 53.3 | 51.4 | 14.7 | 11.1 | 50.20 | 49.17 |
> | 0.3,0.7 | 82.9 | 85.9 | 53.3 | 51.2 | 12.5 | 9.2 | 49.57 | 48.77 |
> | 0.2,0.8 | 83.2 | 86.0 | **53.3** | **52.2** | 10.9 | 7.4 | 49.13 | 48.53 |
> | 0.1,0.9 | **83.4** | **86.6** | 53.2 | 51.7 | 9.9 | 6.7 | 48.83 | 48.33 |
> | 0.0,1.0 | 83.2 | 86.4 | 53.0 | 51.4 | 9.2 | 5.3 | 48.47 | 47.70 |
>
> > **Q3:** Figure 3 lacks the visualization result of the baseline method. It will make audience hard to get the improvement of MDL. The same issue applies to Table 5.
> >
>
> **A3:** Thank you for the suggestion. The baseline method does not employ MDL, so it does not separate specific and shared features and only produces fused features. Therefore, we omit this step from the visualization. Due to rebuttal interface constraints, we cannot upload images here, but we are willing to include this visualization in the revised manuscript.
>
> > **Q4:** It would be helpful to clarify how the 4 testing scenarios under modality-mismatched conditions were selected.
> >
>
> **A4:** NIR is the second most widely used modality in surveillance systems after RGB, and RGB-to-NIR research continues to attract significant interest. Therefore, we evaluate our method in four R-to-N-based application scenarios.

---

### Official Review · Reviewer_Zmjf · 2025-07-02

**Clarity:** 2
**Significance:** 3
**Originality:** 3
**Rating:** 4
**Confidence:** 4

**Summary:**

This paper proposes MDReID framework that addresses any-to-any multi-modal object ReID task, enabling robust re-identification even when query and gallery images come from different modalities. The proposed framework is built on the core idea of decoupling image features into two components: modality-shared features that are consistent across different modalities, and modality-specific features that preserve unique characteristics. This is achieved through two main contributions: Modality-Decoupled Learning (MDL), which uses dedicated tokens in a Vision Transformer to disentangle the features, and Modality-aware Metric Learning (MML), which applies specialized losses to refine the separation. Specifically, MDL extracts modality-shared and -specific features and then compute the robust similarity based on extracted features and availability masks. MML promotes further disentangled & discriminative features through a representation orthogonality loss (ROL) and a knowledge discrepancy loss (KDL). Experiments on three benchmarks show the proposed method approach outperforms state-of-the-art methods, particularly in modality-mismatched scenarios.

**Questions:**

Please address my concerns, especially weaknesses 1, 2, and 3.

**Ethical Concerns:**

["NO or VERY MINOR ethics concerns only"]

**Final Justification:**

Most of my concerns have been addressed during the rebuttal period. Also, I like the potential practical value of the paper. After reading the rebuttal, I'd like to raise my score to 4.

**Limitations:**

yes

**Quality:**

3

**Strengths And Weaknesses:**

**Strengths**

1. The tackled problem, any-to-any multi-modal object ReID, is interesting considering the practicality for real-world applications.
2. Overall, the key idea of the paper is reasonable and the proposed method to achieve this is logical and well-designed (especially ROL & KDL).

**Weaknesses**

1. The comparison to state-of-the-art methods seems potentially unfair due to the use of a "modality availability mask" during inference. This mask provides explicit information about the modality scenario for both the query and gallery, an advantage not available to the baseline methods. This calls into question whether the reported performance gain stems from the novelty of the modality-decoupling approach itself, or simply from leveraging this additional information. To provide a fairer assessment, the paper should compare MDReID against an alternative strategy that also utilizes this mask. For instance, one could train an ensemble of separate "expert" models, each specialized for a specific modality combination (e.g., an R-to-N model, an RT-to-NT model). At inference time, the availability mask could then be used to select and deploy the appropriate expert model for the given scenario. A comparison against such a baseline would better isolate the true contribution of the unified MDReID framework over a simpler, scenario-specific approach.
2. While the paper compares computational costs in terms of parameters and FLOPs, it omits the retrieval cost (i.e., search time), which is a critical metric for ReID systems. The proposed similarity calculation is more complex than a standard feature dot product. Therefore, a comparison of the actual search time against baselines is necessary to fully evaluate the practical efficiency of the proposed method in large-scale gallery settings.
3. To provide stronger intuition for the proposed feature decoupling, a more detailed ablation study is needed. The paper should analyze the individual contributions of the modality-shared and modality-specific features. For instance, what is the performance when using only the shared features or only the specific features during inference? This analysis should be conducted for both modality-matched and mismatched scenarios to clearly demonstrate the respective roles of the decoupled components. Also, the ablation studies of these individual features with different loss functions would be also helpful.
4. (minor) Notations throughout the method is confusing and lacks rigor.

---

> ### Author Rebuttal · Authors · 2025-07-31
>
> Thanks for your constructive feedback. Below, we respond to your key concerns point by point.
> > **Q1:** The comparison to state-of-the-art methods seems potentially unfair due to the use of a "modality availability mask" during inference. This mask provides explicit information about the modality scenario for both the query and gallery, an advantage not available to the baseline methods. This calls into question whether the reported performance gain stems from the novelty of the modality-decoupling approach itself, or simply from leveraging this additional information. To provide a fairer assessment, the paper should compare MDReID against an alternative strategy that also utilizes this mask. For instance, one could train an ensemble of separate "expert" models, each specialized for a specific modality combination (e.g., an R-to-N model, an RT-to-NT model). At inference time, the availability mask could then be used to select and deploy the appropriate expert model for the given scenario. A comparison against such a baseline would better isolate the true contribution of the unified MDReID framework over a simpler, scenario-specific approach.
> >
>
> **A1:** Thank you for your feedback. The use of the modality availability mask is justified, since explicit information about the modality scenario is defaulted as available information and used by all methods. For example, the Complementary Reconstruction Module (CRM) in Top-ReID introduces dense token-level reconstruction constraints to reduce the distribution gap across different image spectra. As a result, it predicts unavailable modality information by leveraging available modalities, representing the current state-of-the-art approach. To this end, TOP-ReID needs to know the explicit information about the modality scenario, and then it can apply the specific generator for reconstruction (e.g., R->T). Meanwhile, following your suggestion, we train four expert models for the RT-to-NT, RT-to-N, R-to-N, and R-to-NT scenarios. Table A shows that our method outperforms these experts by 9.2% on average in mAP and 8.9% in R-1. Furthermore, our single-model approach adapts to all four scenarios, offering greater flexibility. Although Top-ReID also handles multiple scenarios, our method surpasses it by 3.4% in average mAP and 1.1% in average R-1.
>
> Table A: Performance analysis of expert model, Top-ReID, and Ours.
>
> | Scenarios | RT-to-NT |  | RT-to-N |  | R-to-N |  | R-to-NT |  | Average |  |
> | --- | --- | --- | --- | --- | --- | --- | --- | --- | --- | --- |
> |  | mAP | R-1 | mAP | R-1 | mAP | R-1 | mAP | R-1 | mAP | R-1 |
> | Expert model | 37.8 | 36.4 | 4.1 | 2.5 | 3.8 | 1.7 | 4.1 | 1.8 | 12.4 | 10.2 |
> | Top-ReID | 43.0 | 44.6 | 14.4 | 13.3 | 15.4 | **14.0** | 11.9 | 8.7 | 18.2 | 18.0 |
> | Ours | **53.1** | **51.7**  | **16.7** | **13.8** | **16.6** | 11.1 | **15.1** | **10.9** | **21.6** | **19.1** |
>
> > **Q2:** While the paper compares computational costs in terms of parameters and FLOPs, it omits the retrieval cost (i.e., search time), which is a critical metric for ReID systems. The proposed similarity calculation is more complex than a standard feature dot product. Therefore, a comparison of the actual search time against baselines is necessary to fully evaluate the practical efficiency of the proposed method in large-scale gallery settings.
> >
>
> **A2:**  Following your suggestion, we further evaluate the total feature generation and retrieval times of different models on the RGBNT201 test set under the RNT-to-RNT, RT-to-NT, and R-to-N scenarios. Theoretically, our method reduces retrieval cost compared to Top-ReID. For instance, in the RT-to-NT scenario, our approach only matches T-to-T (specific features) and R-to-T (shared features), whereas Top-ReID first generates the missing modality features before performing a full RNT-to-RNT matching. Table B confirms this, showing that our method incurs no additional time and generates features more rapidly.
>
> Table B: Performance analysis under different methods.
>
> | Methods  | Params(M) | Flops(G) | RNT-to-RNT (s) |  | RT-to-NT (s) |  | R-to-N (s) |  |
> | --- | --- | --- | --- | --- | --- | --- | --- | --- |
> |  |  |  | extract | retrieval | extract | retrieval | extract | retrieval |
> | HTT | 85.6  | 33.1 | 8.32 | **0.12** | - | - | - | - |
> | EDITOR | 117.5 | 38.6 | 31.12 | **0.12** | 30.99 | 0.12 | 30.98 | **0.12** |
> | TOP-ReID | 278.2  | 34.5 | 71.78 | **0.12** | 73.32 | **0.11** | 73.71 | **0.12** |
> | Ours  | **57.2**  | **22.3** | **3.27** | 0.14 | **2.90** | **0.11** | **3.14** | **0.12** |
>
> > **Q3:** To provide stronger intuition for the proposed feature decoupling, a more detailed ablation study is needed. The paper should analyze the individual contributions of the modality-shared and modality-specific features. For instance, what is the performance when using only the shared features or only the specific features during inference? This analysis should be conducted for both modality-matched and mismatched scenarios to clearly demonstrate the respective roles of the decoupled components. Also, the ablation studies of these individual features with different loss functions would be also helpful.
> >
>
> **A3:**  Thank you for your suggestion. In our model, both shared and modality-specific features are essential. In the RT-to-NT task, using only shared features overlooks the unique characteristics of the T modality, while using only modality-specific features prevents valid R-to-N matching. To quantify this, we evaluate RT-to-NT performance under three settings: shared features only, modality-specific features only, and both combined. As shown in Table C, combining shared and modality-specific features yields the best performance.
>
> Table C: Performance analysis under different features.
>
> | RT→NT | mAP | R-1 |
> | --- | --- | --- |
> | specific feature | 52.0 | 50.5 |
> | shared feature | 52.7 | 51.2 |
> | specific and shared feature | **53.1** | **51.7** |
>
> > **Q4:** (minor) Notations throughout the method is confusing and lacks rigor.
> >
>
> **A4:** Thank you for the suggestion. We have revised the notations throughout the manuscript to improve readability in recent days. If you have any recommended parts, please provide them in the discussion. We are willing to further clarify them.

---

> > ### Comment · Reviewer_Zmjf · 2025-08-05
> >
> > I appreciate the authors' additional experimental results and discussions. Most of the major concerns I raised have been addressed during the rebuttal period.

---

### Comment · Area_Chair_pU4W · 2025-08-06
**Discussion Required For This Paper**

Dear all,
The authors have submitted their rebuttal. We would appreciate it if you could kindly review their response ASAP and let us know if it affects your assessment or if you have any additional comments. Your input is greatly valued in the decision process.
Even if you entered the mandatory acknowledgement, you also need to offer your comments for the post rebuttal. Thank you again for your time and contribution.
Best,
AC

---

### Decision · Program_Chairs · 2025-09-17

**Decision:**

Accept (spotlight)

**Comment:**

This paper proposes a novel framework, MDReID, for object re-identification across that allows any query and gallery combinations from RGB, NIR, and TIR images, in which the any-to-any object re-identification task is both novel and compelling. The authors make a significant contribution by being the first to accommodate arbitrary query-gallery modality combinations within a unified framework. All reviewers appreciated this paper’s innovation and practicality for real-world applications. The main concerns raised by reviewers were insufficient discussion about KDL, a lack of comparisons of the actual search time against baselines and an inadequately fair assessment. The authors’ rebuttal addressed most of these concerns, resulting in unanimous acceptance by the conclusion of the discussion. Therefore, AC recommends acceptance.